



# A new method to determine the aerosol optical properties from multiple wavelength O₄ absorptions by MAX-DOAS observation

Chengzhi Xing[1], Cheng Liu[1, 2, 3, 7*], Shanshan Wang[4, 5*], Qihou Hu[3], Haoran Liu[1], Wei Tan[3], Wenqiang Zhang[3, 6], Bo Li[1], Jianguo Liu[2, 3]

[1]School of Earth and Space Sciences, University of Science and Technology of China, Hefei, 230026, China
[2]Center for Excellence in Regional Atmospheric Environment, Institute of Urban Environment, Chinese Academy of Sciences, Xiamen, 361021, China
[3]Key Lab of Environmental Optics & Technology, Anhui Institute of Optics and Fine Mechanics, Chinese Academy of Sciences, Hefei, 230031, China
[4]Shanghai Key Laboratory of Atmospheric Particle Pollution and Prevention (LAP³), Department of Environmental Science and Engineering, Fudan University, Shanghai, 200433, China
[5]Shanghai Institute of Eco-Chongming (SIEC), No.3663 Northern Zhongshan Road, Shanghai, 200062, China
[6]School of Environmental Science and Optoelectronic Technology, University of Science and Technology of China, Hefei, 230026, China
[7]Anhui Province Key Laboratory of Polar Environment and Global Change, USTC, Hefei, 230026, China

*Correspondence to*: Shanshan Wang (shanshanwang@fudan.edu.cn) and Cheng Liu (chliu81@ustc.edu.cn)

**Abstract.** Ground based Multi-AXis Differential Optical Absorption Spectroscopy (MAX-DOAS) observation was carried out from November 2016 to February 2017 in Beijing, China to measure the O₄ absorptions in UV and visible bands and

further to illustrate its relationship with aerosol optical properties (AOPs) under different the weather types. According to relative humidity, visibility and PM$_{2.5}$, we classified the observation periods into clear, non-haze, haze, heavy-haze, fog and rainy five different weather conditions. There are obvious differences for measured AOPs under different weather conditions, especially scattering coefficient ($\sigma_{sca}$) and absorption coefficient ($\sigma_{abs}$). It was also found that both the O₄ Differential Slant Column Densities (DSCDs) at UV and visible bands varied in the order of clear days > non-haze days > haze days > heavy-

haze days > fog days. The correlation coefficients ($R^2$) between O₄ DSCDs at 360.8 and 477.1 nm mainly varied in the order of clear days > non-haze days > haze days > heavy-haze days. Based on the statistics of O₄ DSCDs at elevation angle 1° with the corresponding linear regression between UV and visible bands of segmental periods, the relationships between O₄ DSCDs and AOPs were established. It mainly should be clear or non-haze days when the correlation slope is greater than 1.0, correlation coefficient ($R^2$) greater than 0.9 and O₄ DSCDs mainly greater than $2.5 \times 10^{43}$ molec cm$^{-2}$. Meanwhile, $\sigma_{sca}$ and

$\sigma_{abs}$ are less than 45 and 12 Mm$^{-1}$, respectively. For haze or heavy-haze days, the correlation slope is less than 0.6, $R^2$ less than 0.8 and O₄ DSCDs mainly less than $1.3 \times 10^{43}$ molec cm$^{-2}$, under which $\sigma_{sca}$ and $\sigma_{abs}$ are mainly located at 200-900 and 20-60 Mm$^{-1}$. Additionally, the determination method was well validated based on another MAX-DOAS measurement at Gucheng from 19 to 27 November 2016. For more precise and accurate inversion of AOPs, more detailed look-up tables for



O$_4$ multiple wavelength absorptions need to developed. Furthermore, the vertical spatial-resolved aerosol scattering and
absorption information is worthy of being expected by using DSCDs at different elevation angles.

## 1 Introduction

Atmospheric aerosols influence the radiative budget by scattering and absorbing solar radiation directly. It also affects the
global climate change, cloud formation, regional air quality and human health (Seinfeld and Pandis, 2006; Kim and
Ramanathan, 2008; Karanasiou et al., 2012; Levy et al., 2013; Viana et al., 2014). It is important to get a comprehensive
knowledge on the spatial distributions and temporal variations of aerosols and Aerosol Optical Properties (AOPs). Different
aerosols behave obviously different optical properties. For example, Black Carbon (BC) aerosols are characterized by the
strong light absorption. Recent studies indicated that it can heat the air contributing to global warming (Ramanathan et al.,
2007; Galdos et al., 2013; Ramana et al., 2010; Fyfe et al., 2013; Allen et al., 2012). It can also change the atmospheric
temperature vertical profile causing the variations of the planetary boundary layer (PBL) structure (Ding et al., 2016; Wilcox
et al., 2016; Wang et al., 2018). However, dust aerosol and some heterogeneous-reaction secondary aerosols, playing an
important role during the pollution episode in China, are mainly based on scattering optical characteristics. (Huang et al., 2014;
Wang et al., 2018).

Measurements of AOPs, e.g. Aerosol Extinction (AE), Aerosol Optical Depth (AOD), Single Scattering Albedo (SSA),
asymmetry factor and Angstrom, could provide more comprehensive information for a better understanding of the role of
aerosols in atmospheric processes. AOD is an important parameter to evaluate the ability of aerosol particles to attenuate the
solar radiation, which is defined as the integration of AE from surface to the top of atmosphere in vertical. The AE is the sum
of aerosol scattering and absorption coefficients. Moreover, SSA could present the ratio of scattering efficiency to the total
extinction, which is dominant intensive parameter determining aerosols direct radiative forcing. The asymmetric factor is used
to evaluate the aerosol forward scattering ability, while the Angstrom is a parameter to evaluate the aerosol particle size.
Previous measurements of AOPs indicated that the four general aerosol types of biomass burning aerosol, urban-industrial
aerosol, dust aerosol and aerosol of marine origin are exhibiting significant differences in optical properties. The differences
of the optical properties of these kinds of aerosols are used to clarify the mechanisms of aerosol radiative forcing (Dubovik et
al., 2001). For biomass burning aerosol, the Angstrom exponent is mainly distributed between 1.1 and 2.1 at wavelength bands
of 440 – 870 nm and SSA mainly ranging from ~0.88 to 0.99 at 440 nm (Eck et al., 2003; Bergstrom et al., 2007; Weinzierl et
al., 2017). The SSA of urban-industrial aerosol tend to be ~0.95 in cleaner condition and ~0.85 in industrially condition,
respectively (Liousse et al., 1996; Remer and Kaufman, 1998; Garland et al., 2009; He et al., 2009; Shen et al., 2018). Dust
exhibits a pronounced SSA ~0.92 to 0.93 in the blue spectral range at 440 nm, but ~0.96 - 0.99 in longer wavelength greater
than 550 nm (Kaufman et al., 2001; Dubovik et al., 2001; Bergstrom et al., 2007; Weinzierl et al., 2017). The SSA in oceanic
aerosol is mainly greater than 0.97 due to the existing of sea-salt and water soluble particles with high relative humidity (Tanré
et al., 1999; Dubovik et al., 2001; Hess et al., 1998; Eck et al., 2005).



Multi-AXis Differential Optical Absorption Spectroscopy (MAX-DOAS) remote sensing is an effective tool for atmospheric aerosol measurements based on $O_4$ molecular ultraviolet-visible light absorption (Platt and Stutz, 2008). $O_4$ is the collision complex of $O_2$ and its concentration is proportional to the square of the $O_2$ concentration. Due to $O_4$ vertical profile is well known and nearly constant, it can be served as an indicator for the atmospheric distribution photon paths due to its nearly

constant characteristic (Wagner et al., 2004; Frieß et al., 2006; Frieß et al., 2016). The $O_4$ cross-sections exhibit four main absorption bands in the UV-visible region with maxima at 360.8, 477.1, 577.1 and 630.8 nm (Thalman and Volkamer, 2013). By collecting the scattered sunlight spectra at zenith and different elevation angles closed to the horizon by MAX-DOAS, the $O_4$ absorptions can be yielded by the DOAS method and further the aerosol vertical profiles at four different wavelength bands (338-370 nm, 425-490 nm, 540-588 nm and 602-645 nm) (Honninger and Platt, 2002; Hytch et al., 2003; Hönninger et al.,

2004; Wagner et al., 2004; Wittrock et al., 2004; Clémer et al., 2010). The sunlight at different wavelength bands has different ability to traverse the atmosphere, thus the light path length at different wavelength bands are diverse, which can change the corresponding $O_4$ absorptions. Conversely, the correlation analysis between $O_4$ absorptions retrieved at UV range and VIS range could also provide information about the impacts of aerosol scattering on photon paths (Lee et al., 2011). Besides the extinction coefficient profile and AOD, there are no detailed researches on the other AOPs retrieval based on MAX-DOAS

measurements in previous.

In this paper, we try to establish a new method to determine several different aerosol optical properties from multiple wavelength $O_4$ absorptions observed by MAX-DOAS measurement. The measurement of UV and visible $O_4$ absorptions was performed by MAX-DOAS instrument in Beijing from November 2016 to February. Combined the $O_4$ absorptions and measured AOPs, some empirical relationships between them can be found under different weather conditions, which are the

fundamental to determine the AOPs from MAX-DOAS observed $O_4$ absorptions at different wavelength bands. Furthermore, another short period measurement campaign was used to validate the feasibility and reliability of the new method to infer the AOPs under different weather conditions based on the $O_4$ absorptions.

## 2 Measurements and methodology

### 2.1 The MAX-DOAS measurements

The MAX-DOAS instrument was installed on the roof of the Chinese Academy of Meteorological Sciences (CAMS, 39.9475° N, 116.3273° E) for the continuous measurements of $O_4$ absorptions from November 2016 to February 2017. The MAX-DOAS instrument consists of three major parts: a telescope unit, two spectrometers with temperature stabilized at 20° and a computer acting as the controlling and data acquisition unit. The viewing elevation angle of the telescope is controlled by a stepping motor. Scattered sunlight collected by the telescope is redirected by a prism reflector and a quartz fibre bundle to the

spectrometers. Two spectrometers (Acton Spectrapro 300i Czerny-Turner optical spectrometer) were used to cover both the UV (300-460 nm) and visible (400-560 nm) wavelength ranges. The full-width half maximum (FWHM) spectral resolutions of these two spectrometers are all 0.6 nm, or 7.2 detector pixels. Moreover, the optical spectrometer was equipped with a CCD



detector camera (model DU 440-BU) with 2048 pixels. The field of view (FOV) of the instrument is estimated to be less than 0.5º.

A full measurement scanning sequence consists of eleven elevation angles, i.e., 1, 2, 3, 4, 5, 6, 8, 10, 15, 30 and 90°. The instrument azimuth angle is 138° and the exposure time is fixed to 60000 ms for each elevation angle measurement. A full measurement sequence takes about 11 min. Dark current and offset spectra were measured by blocking incoming light using a mechanical shutter and were subtracted from the measurement spectra before spectral analysis. The routine measurements were continuously repeated as long as the Solar Zenith Angle (SZA) was lower than 80°.

**2.2 $O_4$ absorptions in UV and Visible**

The $O_4$ absorptions were derived in the fitting windows of 339 to 387 nm in UV and 425 to 490 nm in visible spectral interval, respectively. The measured spectra were analysed using the QDOAS software developed by BIRA-IASB (http://uv-vis.aeronomie.be/software/QDOAS/). The corresponding zenith spectrum was taken as a reference spectrum for off-zenith elevation angles during each measurement scan. The DOAS fitting generates the Differential Slant Column Density (DSCD)

of $O_4$ between the measured spectra and reference spectrum. Details of DOAS fit settings are listed in Table1. Figure 1 shows a typical DOAS retrieval for the $O_4$ absorptions at 360.8 and 477.1 nm. Afterwards, DOAS fit results with a root mean square (RMS) larger than $5 \times 10^{-4}$ were filtered, and about 99.07% of all $O_4$ measurements remains for the further discussion.

**Table 1. DOAS retrieval settings for $O_4$.**

| Parameter | Data source | $O_4$ Fitting intervals | |
|---|---|---|---|
| | | 338-370 nm | 425-490 nm |
| Wavelength range | | | |
| $NO_2$ | 298K, $I_0$-corrected, Vandaele et al. (1998) | √ | √ |
| $NO_2$ | 220K, $I_0$-corrected, Vandaele et al. (1998) | × | √ |
| $O_3$ | 223K, $I_0$-corrected, Serdyuchenko et al. (2014) | √ | √ |
| $O_3$ | 243K, $I_0$-corrected, Serdyuchenko et al. (2014) | √ | × |
| $O_4$ | 293K, Thalman and Volkamer (2013) | √ | √ |
| HCHO | 298K, Meller and Moortgat (2000) | √ | × |
| H2O | HITEMP (Rothman et al. 2010) | × | √ |
| BrO | 223K, Fleischmann et al. (2004) | √ | × |
| Ring | Calculated with QDOAS | √ | √ |
| Polynomial degree | | Order 5 | Order 4 |
| Intensity offset | | Constant | Constant |

*Solar $I_0$ correction, Aliwell et al., 2002



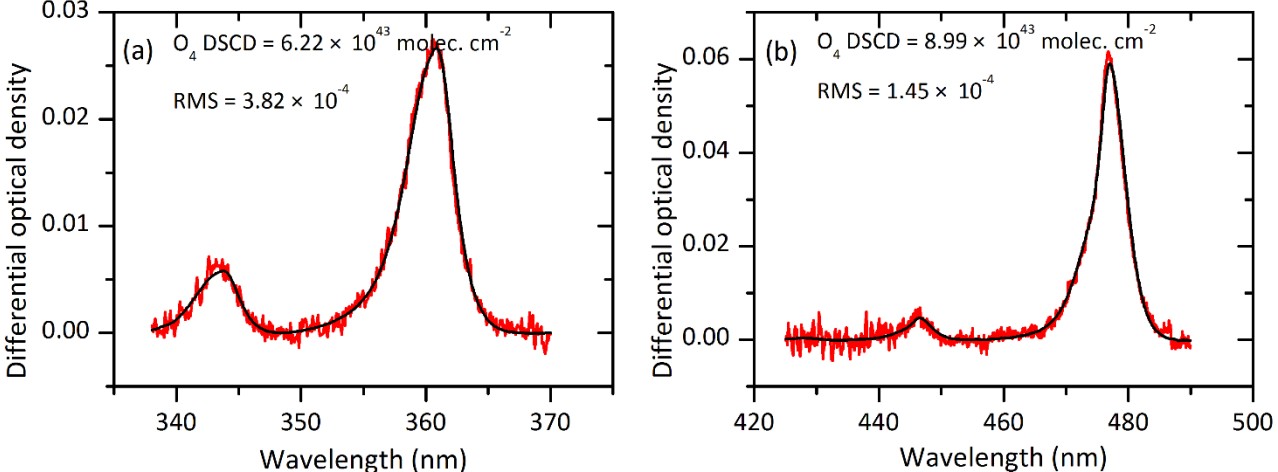

**Figure 1. Typical DOAS spectral fittings for O₄ absorptions in (a) UV and (b) visible bands. Black lines represent the**
**absorption signal and the red lines represent the sum of the absorption signal and the fit residual.**

**2.3 Ancillary data**

Quality-assured Level 2.0 sunphotometer AODs, Asymmetric factor and Angstrom at the Beijing_CAMS AERONET site
(http://aeronet.gsfc.nasa.gov/) were employed, which is collocated with the MAX-DOAS instrument just 2 meters nearby.
Sunphotometer (CE-318) collects direct sunlight several times only during the daytime and only works on non-rainy days.
These aerosol optical parameters at multiple wavelengths were normalized to 450 nm according to Wang et al. (2016). Besides,
the scattering coefficients ($\sigma_{sca}$) were measured at three wavelength ($\lambda$ = 450, 520 and 700 nm) using an integrating
nephelometer (Aurora 4000, Ecotech) at Peking University Urban Atmosphere Environment Monitoring Station (PKUERS,
39.9892° N, 116.3069° E).The absorption coefficients ($\sigma_{abs}$) were measured using a 7-wavelength Aethalometer (AE-31,
Magee Scientific) at $\lambda$ = 370, 470, 520, 660, 880 and 950 nm also located at PKUERS. In order to ensure the accuracy of the
data, the corrections for $\sigma_{sca}$ and $\sigma_{abs}$ were referred to Shen et al. (2018). The SSA was calculated by the measured $\sigma_{sca}$ at
450 nm and estimated $\sigma_{abs}$ at 450 nm using the following equation:

$$SSA = \frac{\sigma_{sca}}{\sigma_{sca} + \sigma_{abs}}$$
(1)

The visibility and the relative humidity (RH) information were collected from the weather history data at Beijing international
airport (http://www.wunderground.com/). In-situ data of PM$_{2.5}$ concentrations were obtained from Guanyuan station (39.9425°
N, 116.3610° E), belonging to the national environmental monitoring network (http://beijingair.sinaapp.com/data/china/sites/),
which is about ~2 km from the CAMS site. All these data are normalized to hourly averages for further discussion.





## 3 Results

### 3.1 Wintertime aerosols optical properties

The time series of PM$_{2.5}$ concentrations, σ$_{sca}$, σ$_{abs}$, SSA, AOD, Angstrom, asymmetry factor and the corresponding
meteorological data, i.e. RH and visibility, from November 2016 to February 2017 are presented in Fig. 2. The typical
meteorological conditions of high RH and low visibility always appeared when PM$_{2.5}$ concentrations increased obviously, and
the corresponding AOD also have a significant growth. As indicated in the gray area of Fig. 2, two episodes of particles
pollution during 15 to 22 December 2016, and 29 December 2016 to 2 January 2017 were identified.




**Figure 2. Time series of PM₂.₅ and AOPs (σ_sca, σ_abs, SSA, AOD, Angstrom and Asymmetry factor), and meteorological parameters (relative humidity and visibility) during the observation in Beijing from November 2016 to February 2017.**





During these two episodes, PM$_{2.5}$ concentrations, $\sigma_{sca}$ and $\sigma_{abs}$ typically increased and remained in a high level for several days, however, decreased faster and declined to a lower value during a shorter while. PM$_{2.5}$ concentrations, $\sigma_{sca}$ and $\sigma_{abs}$ increased to exceeding the maximum values 465 ug cm$^{-3}$, 1331.151 Mm$^{-1}$ and 123.402 Mm$^{-1}$ within 1-3 hours (the increment up to 200 ug cm$^{-3}$, 600 Mm$^{-1}$ and 100 Mm$^{-1}$) during episode I, respectively. In episode II, the maximum values of PM$_{2.5}$ concentrations, $\sigma_{sca}$ and $\sigma_{abs}$ are up to 585 ug cm$^{-3}$, 1473.523 Mm$^{-1}$ and 153.431 Mm$^{-1}$, respectively. In addition, SSA mainly

kept greater than 0.85 during all the wintertime, except it was observed to be less than 0.8 in late November 2016 and several days during January and February 2017. Generally, the high values of SSA were always accompanied by the peak of PM$_{2.5}$ concentrations, which suggests that the scattering properties of atmospheric aerosols were enhanced during the explosive increase stage of particles concentrations. Meanwhile, it is also associated with the decreasing of Angstrom and the increasing of asymmetry factors simultaneously. This is typically related to the particle size growth process (Guo et al., 2014; Yu et al.,

2011; Yu et al., 2016).

   In order to investigate the AOPs under different weather conditions, we classified observation periods of these four months into six scenarios according to the RH, visibility and PM$_{2.5}$ concentration: Clear days (Visibility > 20 km & PM$_{2.5}$ ≤ 35 ug m$^{-3}$ & RH < 80%),Non-haze days (10 km < Visibility ≤ 20 km & 35 ug m$^{-3}$ < PM$_{2.5}$ ≤ 75 ug m$^{-3}$ & RH < 80%), Haze days (RH ≤80% & 5 km < Visibility ≤10 km & 75 ug m$^{-3}$ < PM$_{2.5}$ ≤ 115 ug m$^{-3}$), Heavy-haze days (RH ≤ 80% & Visibility ≤ 5 km &

PM$_{2.5}$ > 115 ug m$^{-3}$), Fog days (RH > 80% & Visibility ≤ 5 km) and Rainy days (Zheng et al., 2015; Duan et al., 2016). As expected, the AOPs showed distinct characteristics during these different weather conditions. Table 2 summarizes the statistics of Air Quality Index (AQI) and AOPs under the six scenarios. AQI is factor to comprehensively evaluate the air quality, which is based on six pollutants of ambient O$_3$, NO$_2$, CO, SO$_2$, PM$_{10}$ and PM$_{2.5}$.

   With the increasing of pollution level indicated by AQI (except fog and rainy days), AOD increased dramatically from 0.311

under clear days to 1.338 in heavy-haze days. There are no obvious changes for $\sigma_{sca}$ and $\sigma_{abs}$ between clear days and non-haze days. Nevertheless, the $\sigma_{sca}$ increased sharply from non-haze days to heavy-haze days with the averaged value from 44.524 Mm$^{-1}$ to 449.741 Mm$^{-1}$. The averaged value of $\sigma_{abs}$ is 8.257 Mm$^{-1}$ in non-haze days and it increased as much as 5 times in heavy-haze days. Moreover, the averaged SSA was about 0.847 on non-haze days and similar to that in haze days, but it increased about 3.53% from haze days to heavy-haze days with the averaged values from 0.846 to 0.878. It suggests that the

aerosol scattering and absorption abilities have changed evidently but the ratio of scattering to extinction have changed slightly during the processes of particle pollution became severe. No obvious variations on Angstrom were observed among clear days to heavy-haze days, but it decreased about 2.83% in fog days. In previous study, the Angstrom are usually higher than 0.80 when AOD is greater than 0.60 in Beijing, which reveals the major contribution of small particles for higher aerosol loading (Che et al., 2015). However, our study demonstrates that small particles made a major contribution to the aerosols throughout

the whole wintertime in Beijing. The obvious decrease of Angstrom in fog days is attributed to the increase of water vapour in particles. In addition, the averaged asymmetry factor was about 0.697 in fog days and 8.52% higher than other weather conditions. It indicates the increased forward scattering in fog days (Yoon and Kim, 2006).



**Table 2. Statistics of AQI and several aerosol optical properties under different weather conditions.**

| Weather condition | Clear day | | | Non-haze day | | | Haze day | | | Heavy-haze day | | | Fog day | | | Rainy day | | |
|---|---|---|---|---|---|---|---|---|---|---|---|---|---|---|---|---|---|---|
| | Visibility > 20 km & PM$_{2.5}$ ≤ 35 ug m$^{-3}$ & RH < 80% | | | 10 km < Visibility ≤ 20 km & 35 ug m$^{-3}$ < PM$_{2.5}$ ≤ 75 ug m$^{-3}$ & RH < 80% | | | 5 km < Visibility ≤ 10 km & 75 ug m$^{-3}$ < PM$_{2.5}$ ≤ 115 ug m$^{-3}$ & RH ≤ 80% | | | Visibility ≤ 5 km & PM$_{2.5}$ > 115 ug m$^{-3}$ & RH ≤ 80% | | | Visibility ≤ 5 km & RH > 80% | | | RH > 80% | | |
| AQI and AOPs | Ave. | min | max | Ave. | min | max | Ave. | min | max | Ave. | min | max | Ave. | min | max | Ave. | min | max |
| AQI | 24 | 5 | 44 | 60 | 15 | 119 | 130 | 39 | 391 | 214 | 43 | 500 | 306 | 26 | 500 | 106 | 15 | 500 |
| σ$_{abs}$ | 7.356 | 0.605 | 63.999 | 8.257 | 1.003 | 37.229 | 39.985 | 2.142 | 103.421 | 53.257 | 3.322 | 105.290 | 89.625 | 7.634 | 156.878 | 28.137 | 2.296 | 94.639 |
| σ$_{sca}$ | 41.411 | 3.920 | 214.581 | 44.524 | 8.889 | 305.853 | 259.081 | 5.872 | 809.550 | 449.741 | 14.093 | 1096.859 | 739.152 | 53.895 | 1662.896 | 217.125 | 25.938 | 656.143 |
| SSA | 0.854 | 0.419 | 0.975 | 0.847 | 0.518 | 0.953 | 0.846 | 0.438 | 0.931 | 0.878 | 0.686 | 0.930 | 0.887 | 0.790 | 0.928 | 0.878 | 0.764 | 0.941 |
| Asy | 0.640 | 0.560 | 0.714 | 0.643 | 0.599 | 0.670 | 0.639 | 0.575 | 0.704 | 0.647 | 0.598 | 0.742 | 0.697 | 0.660 | 0.708 | | | |
| Angstrom | 1.252 | 0.210 | 1.943 | 1.304 | 0.429 | 1.950 | 1.265 | 0.176 | 1.853 | 1.286 | 0.798 | 1.731 | 1.054 | 0.568 | 1.759 | | | |
| AOD | 0.311 | 0.051 | 0.799 | 0.351 | 0.103 | 0.927 | 0.892 | 0.645 | 2.495 | 1.338 | 0.939 | 2.693 | 0.998 | 0.105 | 2.509 | | | |






## 3.2 UV and visible O₄ absorptions under different weather conditions

Figure. 3 shows the examples of diurnal pattern and corresponding correlation of UV and visible $O_4$ DSCDs (elevation angle
= 1°) at 360.8 and 477.1 nm under five different weather conditions, except for the rainy days. In view of the absolute strength
of $O_4$ absorption, both the $O_4$ DSCDs at UV and visible bands varied in the order of clear days > non-haze days > haze days >

heavy-haze days > fog days. It manifested the dependency of $O_4$ absorption on the scattering sunlight path impacted by the
aerosol loading. Moreover, $O_4$ DSCDs at 477.1 nm are obviously higher than that at 360.8 nm in clear and non-haze days, and
slightly larger than that at 360.8 nm in haze and heavy-haze days, which can be explained by the fact that the observable light
path length at visible range is longer than UV range. Even in UV bands, the observed $O_4$ DSCDs at 353 nm were reported to
be lower than those at 380 nm for most of the elevations under haze conditions during winter in Beijing (Lee et al. 2011). This

phenomenon revealed that $O_4$ absorptions in short wavelength range were more significantly affected by light diffusion under
hazy conditions. However, we found there is no obvious differences between $O_4$ DSCDs at 360.8 and 477.1 nm in fog days,
during which the high contents of water vapour decreased the visibility and the atmospheric absorption paths from UV to
visible range.

We further analysed the relationship of $O_4$ absorptions between UV and visible bands. As shown in the right column of Fig. 3,

the correlation coefficient ($R^2$) of $O_4$ DSCDs between at 360.8 and 477.1 nm varied in the order of clear days > non-haze days >
haze days > heavy-haze days. Strong correlation between UV and visible $O_4$ absorptions ($R^2 > 0.9$) was achieved for clear and
non-haze days. Under haze and heavy-haze conditions, $R^2$ was 0.81 and 0.74, respectively, which are much lower than that in
clear and non-haze days. That is because the increase of light-absorbing and light-scattering aerosols can result in reduced
light path lengths more obviously in shorter wavelength bands than longer wavelength bands during haze and heavy-haze days.






**Figure 3. Diurnal variation and correlation analysis of O₄ DSCDs at 360.8 and 477.1 nm under different weather conditions: (a) and (b) clear day, (c) and (d) non-haze day, (e) and (f) haze day, (g) and (h) heavy-haze day, (i) and (j) fog day. The colorbar represents the time sequence.**





The changes of AOPs, especially aerosol scattering and absorption properties, are mainly manifested in the variations of $O_4$ absorptions at different wavelength bands. The correlation information of $O_4$ DSCDs at different bands will also be affected by the variation of AOPs. For more detailed, i.e., 21 February 2017, was chosen to exhibit the influence of AOPs changes on $O_4$ DSCDs in Fig. 4. Compared Fig. 4 (a) to (b), it can be found that $\sigma_{sca}$ and $\sigma_{abs}$ have a similar variation trends, a slightly

turning and an abruptly decrease occurred at ~09:05 and ~12:00 (especially for $\sigma_{sca}$), respectively, while the time-indicated $O_4$ DSCDs seems to be three segments with higher correlation coefficient divided by the break point of 10:00 and 12:00.

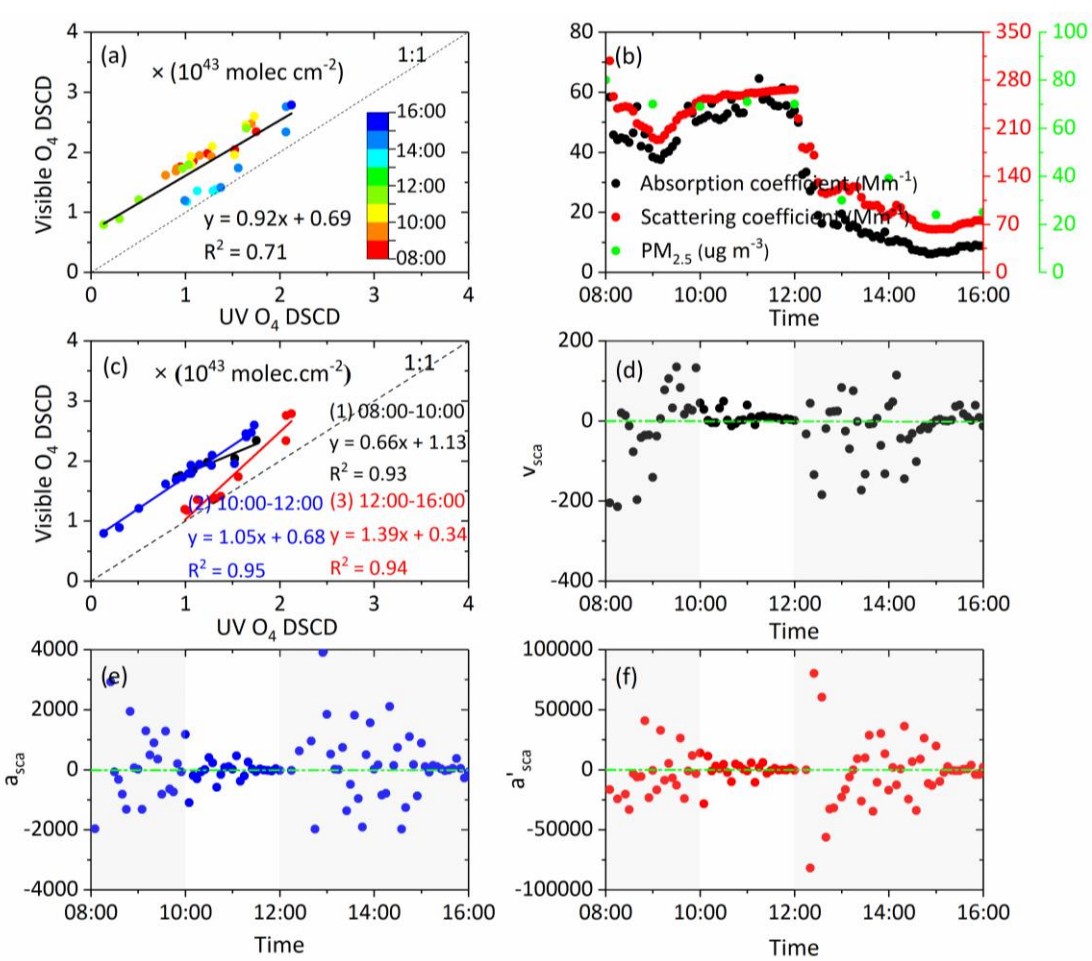

**Figure 4.** An example day on 21 February 2017: (a) the correlations between $O_4$ DSCDs at 360.8 and 477.1 nm. The
colorbar represents time sequence. (b) shows the time series of aerosol scattering and absorption coefficients. The
correlations information between $O_4$ DSCDs at 360.8 and 477.1 nm on 08:00-10:00, 10:00-12:00 and 12:00-16:00 21
February 2017 were shown in (c). (d) to (f) shows the time series of $v_{sca}$, $a_{sca}$ and $a'_{sca}$ of scattering coefficients,
respectively.





In order to explore the relationship between the $O_4$ DSCDs at different wavelength bands and the variations of $\sigma_{sca}$ and $\sigma_{abs}$, we defined the change speed ($v_{sca}$), acceleration ($a_{sca}$) and the change rate of acceleration ($a'_{sca}$) of $\sigma_{sca}$ (Fig4 (b), (e) and (f)) as the following three equations,

$$v_{sca} = \frac{d_{sca}}{dt} \tag{2}$$

$$a_{sca} = \frac{d_{v_{sca}}}{dt} \tag{3}$$

$$a'_{sca} = \frac{d_{a_{sca}}}{dt} \tag{4}$$

Accordingly, the relevant time series of $v_{sca}$, $a_{sca}$ and $a'_{sca}$ are displayed in Fig. 4 (d) to (f). In this case, we can find two time break points, defined as $t_1$ and $t_2$ ($t_1$ = 10:00 and $t_2$ = 12:00), at which $\sigma_{sca}$ and $\sigma_{abs}$ have significant variations based on the calculated $v_{sca}$, $a_{sca}$ and $a'_{sca}$. In addition, we found the indicator of $a'_{sca}$ can describe the specific moment at which the change (increasing or decreasing) of $\sigma_{sca}$ more clearly than $v_{sca}$ and $a_{sca}$ in this case. | $a'_{scat_1}$ | and | $a'_{scat_2}$ | are all higher than 20000.

Consequently, the $O_4$ DSCDs at 360.8 and 477.1 nm can be divided into three segments for the periods of 08:00-10:00, 10:00-12:00 and 12:00-16:00 and the correlation between UV and VIS $O_4$ DSCDs was further analysed individually. As shown in Fig. 4(c), the $R^2$ during 08:00-10:00 and 10:00-12:00 is obviously larger than that of all day in Fig. 4(a), however, it is smaller for segment of 12:00-16:00. Moreover, there were huge divergences among the correlation slopes among these three segments due to the change of aerosol scattering and absorption properties. Therefore, it can be concluded that the diurnal variations of

$O_4$ DSCDs provide the information of the light path length impacted by aerosol loading, and further the varied relationship between $O_4$ DSCDs at UV and visible implies the change of the aerosol scattering and absorption properties.

Using the method discussed above, we have defined the time break points with aerosol properties changes and further classified the observation into several segmental periods with the criterion of $|v_{sca}| > 1000$ or $|a_{sca}| > 10000$ or $|a'_{sca}| > 20000$. The summary of time break points and corresponding change speed ($v_{sca}$), acceleration ($a_{sca}$) and the change rate of acceleration

($a'_{sca}$) of $\sigma_{sca}$ were listed in Table S1.

### 3. 3 Implications of $O_4$ absorptions to aerosol optical properties

In order to derive the aerosol optical properties from multiple wavelength $O_4$ absorptions, the complete four months observational $O_4$ and AOPs data were used for discussion under different weather types. Hourly data of $O_4$ DSCDs were divided into five weather conditions and made the linear regression between UV and VIS $O_4$ DSCDs. In total, there were about

218 segments (776 hours in 97 days), including 67, 31, 61, 44 and 15 segments for clear, non-haze, haze, heavy-haze and fog days, respectively. Figure 5 illustrated the statistics of $O_4$ DSCDs in UV and visible bands, and the slope and $R^2$ of correlation analysis between them, as well as the $O_4$ DSCDs ratio of UV to visible for different weather conditions.





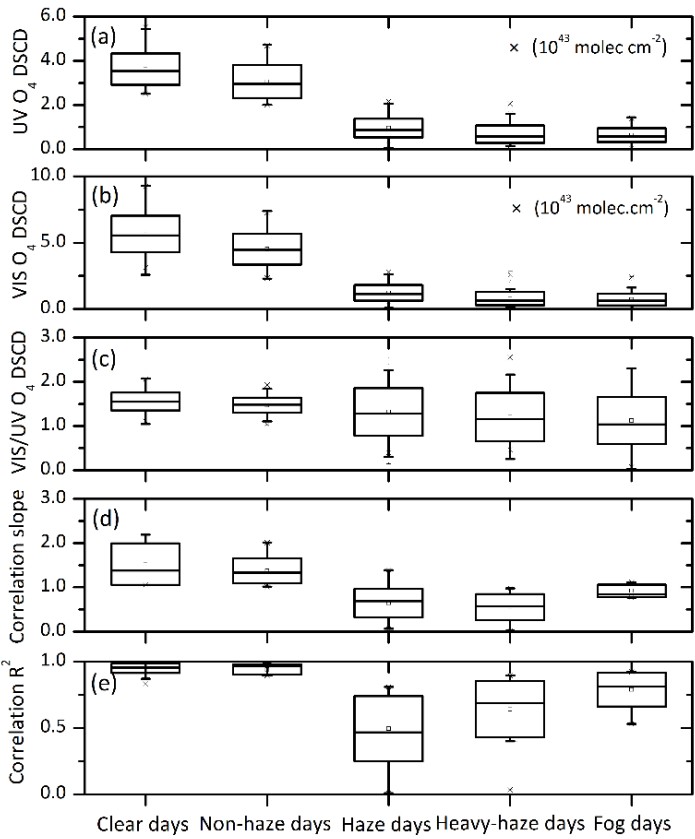


**Figure 5. Box plots of statistics on the O₄ DSCD under different weather conditions: (a) at UV band, (b) at visible band, (c) the ratio of VIS/UV O₄ DSCD, (d) correlation slope and (e) $R^2$ between UV and visible (Visible O₄ DSCD=slope*UV O₄ DSCD + intercept).**

In general, the O₄ DSCDs in UV are mainly ranged in 3.00-4.00 × $10^{43}$, 2.50-3.50 × $10^{43}$, 0.50-1.10 × $10^{43}$, 0.25-0.80 × $10^{43}$

and 0.20-0.40 × $10^{43}$ molec cm⁻² in clear, non-haze, haze, heavy-haze and fog days, respectively. And the O₄ DSCDs in visible are mainly distributed between 4.00-6.50 × $10^{43}$, 3.00-5.50 × $10^{43}$, 0.50-1.30 × $10^{43}$, 0.25-0.60 × $10^{43}$ and 0.25-0.60 × $10^{43}$ molec cm⁻² under above five different weather conditions, which are higher than those in UV especially for clear and non-haze days. Moreover, the corresponding ratio of visible to UV O₄ DSCDs are 1.45-1.70, 1.45-1.65, 1.00-1.65, 0.85-1.35 and 0.80-1.35 under these five weather conditions, respectively. The linear regression results show that the correlation slopes between

UV and visible O₄ DSCDs are greater than 1.00 (mainly greater than 1.40) and the correlation $R^2$ are greater than 0.93 mostly in clear days. Under non-haze condition, the correlation slopes are greater than 1.00 (mainly greater than 1.20) and the correlation $R^2$ are mainly greater than 0.90, respectively. The correlation slopes are mainly less than 0.60 and the correlation $R^2$ have a wider range (the maximum value < 0.80 and occasional fitting failure) in haze days. In heavy-haze days, the correlation slopes are less than 0.60-0.80 and the correlation $R^2$ are 0.50-0.80 mostly (some fitting failure cases appeared). In

fog days, the correlation slopes are floated around 1.00 and the correlation $R^2$ are mainly 0.75-0.85, respectively.





Meanwhile, the statistical characteristics of AOPs under different weather conditions are shown in Fig. 6. Similar to the results in Table 2, both $\sigma_{sca}$ and $\sigma_{abs}$ show the increasing trend in clear, non-haze, haze, heavy-haze and fog days, which were mainly distributed between 20-50, 30-60, 130-350, 230-650, 450-1000 Mm$^{-1}$ and 3-8, 4-12, 20-60, 35-70, 70-115 Mm$^{-1}$, respectively. The AODs in clear and non-haze days were mainly distributed between 0.1-0.35, and significantly increased to 0.8-2.4 in haze
and heavy haze days. The Angstrom were more disperse for clear and non-haze days than the other conditions. Except the fog days, the asymmetry factor in other weather conditions are not much different.

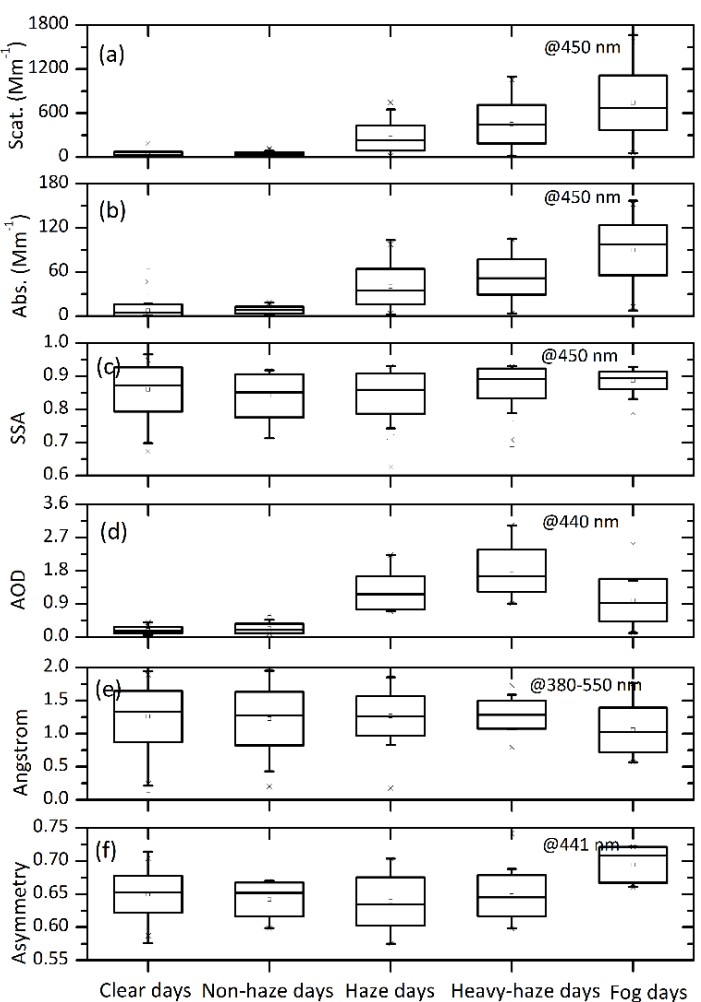

**Figure 6. Box plots of the statistics on aerosol optical properties under different weather conditions: (a) $\sigma_{sca}$, (b) $\sigma_{abs}$,**
**(c) SSA, (d) AOD, (e) Angstrom and (f) Asymmetry.**





Combined the statistical information on $O_4$ absorptions and AOPs, we could conclude some empirical relationships as following:

(1) Under the condition that the correlation slopes between UV and visible $O_4$ DSCDs greater than 1.0 and the correlation $R^2$

greater than 0.9, simultaneously, the UV and visible $O_4$ DSCDs are mainly greater than $2.5 \times 10^{43}$ molec cm$^{-2}$ and $3.0 \times 10^{43}$ molec cm$^{-2}$, we could know the weather mainly should be clear or non-haze days. It can be suspected that $\sigma_{sca}$ and $\sigma_{abs}$ are less than 45 Mm$^{-1}$ and 12 Mm$^{-1}$, and AODs are less than 0.4.

(2) Under the condition of the correlation slope less than 0.6 and the correlation $R^2$ less than 0.8, simultaneously, the UV and visible $O_4$ DSCDs are mainly less than $1.1 \times 10^{43}$ molec cm$^{-2}$ and $1.3 \times 10^{43}$ molec cm$^{-2}$, the weather mainly should be

haze or heavy-haze days. Moreover, $\sigma_{sca}$ and $\sigma_{abs}$ are estimated to be distributed at 200-900 Mm$^{-1}$ and 20-60 Mm$^{-1}$, respectively. AODs are between 0.9 and 2.5. In more detail, $\sigma_{sca}$, $\sigma_{abs}$ and AOD will be located at 200-400 Mm$^{-1}$, 20-50 Mm$^{-1}$ and 0.9-1.5 under the condition of UV and visible $O_4$ DSCDs $> 1.0 \times 10^{43}$ molec.cm$^{-2}$.

(3) If the correlation slope floating around 1.0 and with a correlation $R^2$ of 0.75-0.85, we could know the weather mainly should be fog days. $\sigma_{sca}$ and $\sigma_{abs}$ are located at 450-1200 Mm$^{-1}$ and 60-90 Mm$^{-1}$, while AODs are greater than 0.7.

Therefore, it represents the potential ability to determine the basic aerosol loading conditions from the MAX-DOAS observed $O_4$ absorptions.

## 4 Discussion

To investigate the feasibility and reliability, another short period MAX-DOAS measurement campaign operated in Gucheng, Hebei province (39.1382° N, 115.7163° E) from 19 to 27 November 2016 was used for the application of the new method to

determine AOPs from $O_4$ absorptions. The MAX-DOAS instrument is the same as that one installed in CAMS. Due to absence of sunphotometer instrument, AODs at 450 nm were obtained by profiling the aerosol extinction coefficient based on MAX-DOAS measurements by utilizing the optimal estimation method (Frieß et al., 2006; Frieß et al., 2016; Xing et al., 2017). Besides, $\sigma_{sca}$ and $\sigma_{abs}$ were acquired using the co-located same integrating nephelometer (Aurora 1000, Ecotech) and 7-wavelength Aethalometer (AE-31, Magee Scientific), respectively.

Figure 7(a) and (b) shows the diurnal variations and segmental correlation of $O_4$ DSCDs in UV and visible bands during this campaign. According to the empirical relationships discussed in section 3.3, it can be inferred that the period segment during 09:00-11:00 in 25 November should be haze or heavy-haze weather type, because the UV and VIS $O_4$ DSCDs are all less than $0.5 \times 10^{43}$ molec cm$^{-2}$, and simultaneously the correlation slope and $R^2$ between them are 0.42 and 0.59, which are in line with the determination conditions that UV and visible $O_4$ DSCDs are mainly less than $1.1 \times 10^{43}$ and $1.3 \times 10^{43}$ molec cm$^{-2}$,

simultaneously combined the correlation slope and $R^2$ between them are mainly less than 0.6 and 0.8. Similarly, other periods that 09:00-12:00 of 21 Nov., 10:50-16:00 of 22 Nov., 10:00-15:00 of 23 Nov., 08:00-15:00 of 26 Nov. and 11:00-15:00 of 27 Nov. are mainly clear or non-haze weather type. And 09:00-10:00 of 19 Nov., 09:00-12:00 of 20 Nov. and 09:00-10:50 of 22 Nov. can be mainly regarded as haze or heavy-haze weather types.





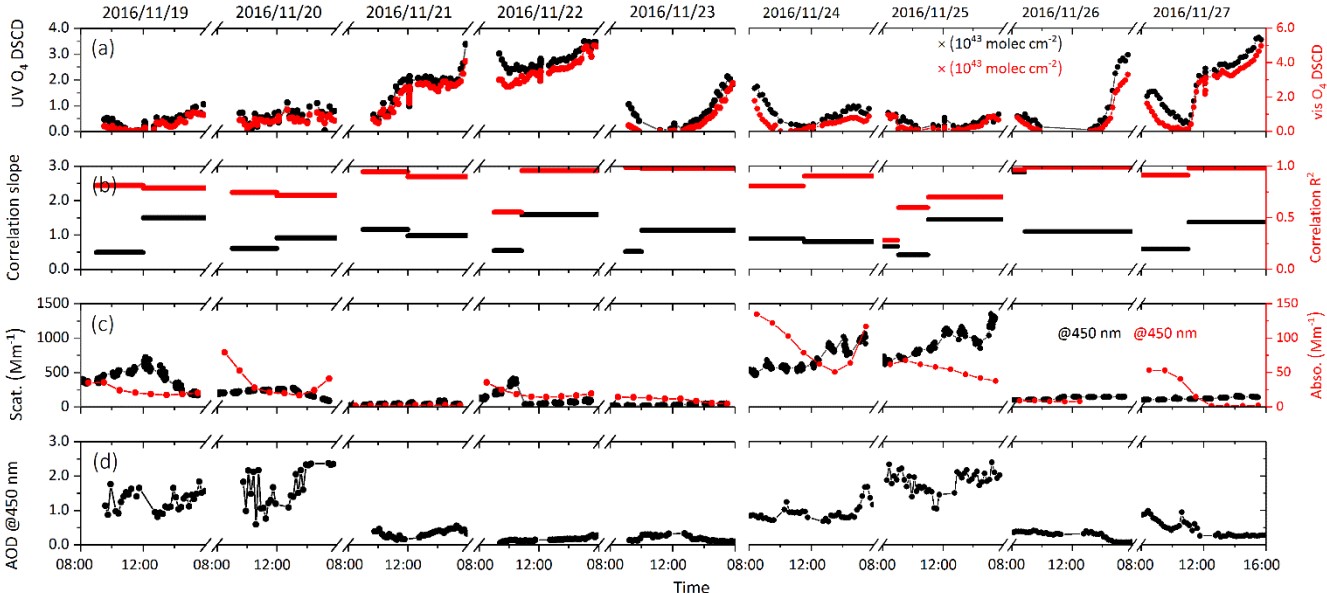


**Figure 7. Time series of O₄ absorptions and aerosol optical properties at Gucheng, Hebei from 19 to 27 November 2016: (a) UV and visible O₄ DSCDs, (b) correlation slopes and R² between O₄ DSCDs at 360.8 and 477.1 nm, (c) $\sigma_{sca}$ and $\sigma_{abs}$ at 450 nm, (d) AOD at 450 nm retrieved by MAX-DOAS.**

Furthermore, the time series of in-situ $\sigma_{sca}$, $\sigma_{abs}$ and MAX-DOAS retrieved AOD are shown in Fig.7 (c) and (d), which are helpful to validate the AOPs determined according to the empirical relationships summarized above. The $\sigma_{sca}$, $\sigma_{abs}$ and AOD are mainly located at 200-900 Mm⁻¹, 20-60 Mm⁻¹ and 0.9-2.5 under haze or heavy-haze conditions, respectively. The in-situ $\sigma_{sca}$, $\sigma_{abs}$ and MAX-DOAS retrieved AOD of the identified haze segment of 09:00-11:00 of 25 November are ranged in 588.30-730.77 Mm⁻¹, 58.19-67.63 Mm⁻¹ and 1.39-2.22. It indicates that the concluded empirical relationships can be used as 

the criterion to accurately determine the ranges of aerosol optical parameters of $\sigma_{sca}$, $\sigma_{abs}$ and AOD. Nevertheless, we found two segments with correlation slopes > 1.0 and R² < 0.9 during 12:00-15:00 of 19 Nov. and 11:00-15:00 of 25 Nov., which is not included in cases of the empirical relationships. It suggests that more refined and quantitative relationships between aerosol optical parameters and O₄ absorptions need to be further achieved with the increases of the measured data, which can be established as a look up table to retrieve the aerosol optical properties in the future.

**5 Summary and conclusions**

Ground-based MAX-DOAS measurements for O₄ DSCDs at UV and visible wavelength bands were carried out in Beijing from November 2016 to February 2017. Combined with the measured $\sigma_{sca}$ and $\sigma_{abs}$ and AOD, we have summarized the

characteristics of $O_4$ absorptions and parameters of AOPs under different weather conditions during autumn-winter seasons. It was found that the averaged AOD increased from 0.311 in clear days to 1.338 in heavy-haze days. The averaged $\sigma_{sca}$ changed dramatically from 44.524 $Mm^{-1}$ in non-haze days to 449.741 $Mm^{-1}$ in heavy-haze days. Moreover, the averaged $\sigma_{abs}$ also obviously increased from 8.257 $Mm^{-1}$ in non-haze days to 53.257 $Mm^{-1}$ of heavy-haze days to. Both the measured UV and visible $O_4$ DSCDs varied in the order of clear days > non-haze days > haze days > heavy-haze days > fog days. The corresponding correlation information (slope and $R^2$) between $O_4$ DSCDs at UV and visible wavelength bands also changed synchronously when $\sigma_{sca}$ and $\sigma_{abs}$ have varied.

Considering the simultaneous variation of $O_4$ absorptions and AOPs, the segmental periods correlations analysis between UV and visible $O_4$ DSCDs were performed. Afterwards, the empirical relationships between $O_4$ absorptions and AOPs can be concluded for different aerosol loadings. It could be clear and non-haze days under the condition of correlation slopes are greater than 1.0 and $R^2$ mainly greater than 0.9, simultaneously, UV and Visible $O_4$ DSCDs are mainly greater than $2.5 \times 10^{43}$ molec $cm^{-2}$. $\sigma_{sca}$, $\sigma_{abs}$ and AOD are mainly less than 45 $Mm^{-1}$, 12 $Mm^{-1}$ and 0.4 under this condition, respectively. When the correlation slopes, $R^2$ and $O_4$ DSCDs are less than 0.6, 0.8 and $1.3 \times 10^{43}$ molec $cm^{-2}$, respectively, it mainly should be haze or heavy-haze days. Under this condition, the $\sigma_{sca}$, $\sigma_{abs}$ and AOD can be inferred to be mainly located at 200-900 $Mm^{-1}$, 20-60 $Mm^{-1}$ and 0.9-2.5, respectively. In addition, it should be fog days if the correlation slopes float around 1.0 and $R^2$ of 0.75-0.85. Another MAX-DOAS measurement campaign carried out at Gucheng from 19 to 27 November 2016 were used to validate the proposed new method, which could well determine the AOPs from the observed $O_4$ absorptions.

In this paper, we present a new method to deduce directly the parameters of aerosol optical properties from the observed UV and visible $O_4$ absorptions, which expands the usages of MAX-DOAS technique to fast semi-quantify the aerosol scattering and absorption properties. With the improvement of the look-up table, more precise and accurate inversion of aerosol optical properties can be achieved. Since only the $O_4$ DSCDs at elevation angle of 1° were employed to obtain the aerosol scattering and absorption at surface, it can be expected that vertical spatial-resolved of aerosol scattering and absorption can be retrieved by using $O_4$ DSCDs at different elevation angles in the future study.

**Data availability.** Data are available for scientific purposes upon request to the corresponding authors.

**Author contributions.** CX, CL and SW designed and implemented the research, as well as prepared the manuscript; QH contributed to analysis of the in-situ AOPs data; CX, HL and WT carried out the MAX-DOAS observations in CAMS and Gucheng sites; WZ and BL contributed to the MAX-DOAS data retrieval; JL provided constructive comments on this research.

**Competing interests.** The authors declare that they have no conflict of interest.

...





**Acknowledgements**

This research was supported by grants from National Key Research and Development Program of China (2018YFC0213104, 2018YFC0213100, 2016YFC0203302, 2017YFC0210002), National Natural Science Foundation of China (41722501, 91544212, 51778596, 41575021), and Shanghai Pujiang Talent Program (17PJC015). We would like to thank CAMS and Peking University for the data of $\sigma_{sca}$ and $\sigma_{abs}$ measured in Gucheng and PKUERS, respectively.

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
