# Peer review of "A new method to determine the aerosol optical properties from multiple wavelength O4 absorptions by MAX-DOAS observation"

_Atmospheric Measurement Techniques, 2019_

## Referee Comment (RC1) · Anonymous Referee #1 · 22 Feb 2019

The paper entitled "A new method to determine the aerosol optical properties from multiple wavelength O4 absorptions by MAX-DOAS observation" by Xing et al. presents a new method to determine the multiple aerosol optical properties (AOPs) from the MAX-DOAS observation of O4 absorption at UV and visible wavelength during the autumn-winter seasons. According to the RH, visibility and PM2.5 concentrations, the observation periods were classified into different types, among which the AOPs behave significantly different. Then the empirical relationships between measured O4 absorptions in different bands and characteristics of AOPs were summarized and further used as a new method to determine the aerosol optical properties, which is well validated in another independent campaign. The manuscript is generally well written,

clearly presented and is recommended for publication in AMT after some minor corrections. Major concern: The new method is based on the dependence of O4 absorptions in different wavelength bands on the aerosol optical properties, which is summarized from the measurements. So I suggest the authors can present some evidences of theoretical estimation with the forward RTM to enhanced the principle basis of the method somewhere, even in the supplementary materials. Minor comments: P2, L34: need to be developed P2, L83, February of which year? P4, Sect. 2.2 & P4, Fig. 1: Please provide the basic information of the measured spectrum in the fitting example, which can help to evaluate the performance of spectral analysis better. P6, L143: growth -> increase P6, L161-165: Any special consideration for using different RH for clear days, non-haze days (RH<80%) with haze days (RH$\leq$80%)? P10, Fig. 3: Besides the discussion about the correlation coefficient, could the authors give some explanations of the changes in slopes among different weather types? Obviously, the slope in clear and non-haze days are much larger than those in haze and heavy-haze days. Why? P12, Fig. 4: The scat. and abs. changed around 09:05 and 12:00, while the correlation relationship analysis use the break point of 10:00 and 12:00. Why they are different in time? Moreover, why the authors choose the index of variations of scat. instead of abs.? P13-14, Fig. 5, Fig. 6: the empirical relationships between measured O4 absorptions in different bands and characteristics of AOPs were mainly concluded from the statistic plots of Fig. 5 and Fig. 6. I have a concern that the some of the factors (e.g. correlation R2 and VIS/UV O4 DSCD in haze days, as well as scat. and abs.) have wide value range even covers some cases of other weather conditions. How to obtain the precise and accurate the correspondence between O4 absorptions and AOPs under different weather conditions? P16-17, Sect. 4: For the validation, the authors classified the observational period segments into the different weather conditions, however, no further AOPs information, e.g. AODs, sca. and abs., were inferred and achieved. Is the sentence in line 321 ("The $\sigma$sca, $\sigma$abs and AOD are mainly located at 200-900 Mm-1, 20-60 Mm-1 and 0.9-2.5 under haze or heavy-haze conditions, respectively.") a conclusion of measurement results or inference from O4 absorptions?

---

## Referee Comment (RC2) · Anonymous Referee #2 · 6 Mar 2019

This paper presents a comparative analysis of O4 absorptions from MAX-DOAS observations and aerosol optical properties from a ground-based sun photometer and in-situ measurements. Then, the co-variation between the O4 absorption and aerosol properties was used to determine the aerosol conditions (clear, non-haze, haze, heavy-haze, etc). This work addresses an interesting topic. But, the concept of this study has some major flaws as I listed below. I don't recommend for publication before those major concerns are addressed.

Major criticism:

1. The approach to infer aerosol loading condition from the O4 absorptions is too

far from quantitive, although authors mentioned to develop a lookup table to retrieve aerosol optical properties in the future. Aerosol loading information in the level of conditions defined in this study (clear, non-haze, haze, heavy-haze, etc) can be easily told from human eyes. Therefore, the aerosol information inferred from the method of this study has no scientific value.

2. Physically, the study methodology is not as much as sound. The O4 absorption derived from the MAX-DOAS observations are indeed contaminated by aerosols. So in their method, authors intend to use an aerosol-impacted O4 absorption data to infer the aerosol properties. I think a better way to get information from the MAX-DOAS observation is to retrieve aerosol information (if there is aerosol information) along with the O4 absorption. Or from another perspective, to study the impact of aerosols to the retrieval accuracy in the O4 absorption.

3. I am not convinced with the use of change speed, acceleration, and change rate of the acceleration of diurnal scattering coefficient (equations 2-4) for describing the aerosol property change. These variables only make the trivial diurnal analysis more complicated.

4. The presentation quality needs improvement. This paper has many grammar issues. I try to catch them in the technical comments below.

Specific comments:

1. It seems different datasets have different sampling times. It is not clear based on what time length the data are aggregated and compared. Please clarify this.

2. Line 160-165: the weather conditions are called clear DAYS, . . . Rainy DAYS. Does it mean that all the data are analysis with on a daily basis? If yes, it may be not appropriate, because different weather conditions can happen within a day. If no, these categories should not be called xx Days.

3. I would suggest change "non-haze" into "light-haze", as "non-haze" means clear.
And this condition has an average AOD of 0.35; calling light-haze is more proper.

4. Why the elevation angle = 1 degree is chosen for O4 DSCD used in this study? Please clarify it in the article.

Technical corrections:

Line 24: I don't understand the meaning of "O4 Differential Slant Column Densities (DSCDs) at UV and visible bands varied in the order of ....". Do you mean the magnitude of their correlation coefficients decrease in the order of ...?

40: aerosols and Aerosol Optical -> aerosol loading and Aerosol Optical

40-41: Different aerosols behave -> Different aerosol types behave

43: heat the air contributing to -> heat the air, and contributes to

44: profile causing -> profile, causing

48: Please note that AE often refers to Angstrom Exponent.

49: Angstrom -> Angstrom Exponent

51: atmosphere in vertical. -> atmosphere.

52: SSA could represent -> SSA is defined as

55: the four general aerosol types of biomass burning aerosol, urban-industrial aerosol, dust aerosol and aerosol of marine origin are exhibiting -> different aerosol types (such as biomass burning, urban-industrial, dust, and sea-salt aerosols) exhibit

63: pronounced SSA -> SSA

71: with maxima at -> around

73: O4 absorptions can be yielded by the DOAS method and further the aerosol vertical profiles at four different wavelength bands (xx). -> O4 absorptions in four bands (xx, xx, xx, and xxmm ) can be estimated, and aerosol vertical profiles can be further derived.

[Figure]

90: Science (CAMS -> Science building (CAMS

95: were used to cover -> are used to cover

93: was equipped with -> is equipped with

106: in UV -> in the UV; in visible spectral interval -> in the visible

113: filtered -> filtered out; measurements remains for the further discussion. -> measurements remained.

Table 1: The second line is confusing; No asterisk mark can be found for the table footnote.

124: several times only during the daytime and only works on non-rainy days-> about every 15 minutes during non-rainy daytime.

133: Please indicate the distance between Beijing Airport and your site.

141: always appeared when PM2.5 concentrations increased obviously, and the corresponding AOD also have a significant growth. -> coincided with significantly high PM25 concentration and high AOD.

143: have a significant growth -> increases dramatically

143: gray area -> gray areas

143: particles pollution -> particulate pollution

150: decreased faster and declined to -> decreased sharply to

150: during a shorter while -> within a shorter period

153: are up to -> are

154: greater than -> over; all the wintertime -> the entire wintertime

176: decreased about -> decreased by about

189: at UV -> in the UV

Figure 3: please indicate the time is Beijing Time (UTC+8)

248: weather types -> weather conditions

255: on the O4 -> for the O4

255: at UV band -> in the UV band; at visible -> in the visible

306: weather type -> condition

325: table. to retrieve -> table to retrieve

330: at UV and visible wavelength bands -> in the UV and visible bands

335: heavy-haze days to. -> heavy-haze condition.

341: correlation slope -> linear-regression slope

---

## Author Comment (AC1) · 28 Apr 2019

First of all, we would like to thank the reviewer for this positive assessment of our manuscript, the constructive and helpful suggestions. Point-to-point responses are given below. The original comments are black in color, while our responses are in blue. All the mentioned line number are referred to the revised manuscript.

Major comments

The new method is based on the aerosol optical properties, which is summarized from the measurements. So I suggested the authors can present some evidences of theoretical estimation with the forward RTM to enhanced the principle basis of the method somewhere, even in the supplementary materials.

R: Thanks for your great suggestion. In order to enhance the principle basis of this method described in the manuscript, we used radiative transfer model of SCIATRAN to simulate $O_4$ DSCDs in UV and Visible bands under conditions with different aerosol optical properties. As listed in Table R1, 11 different aerosol scenarios were simulated in total, in which case 1 is the default case to represent haze condition. Case 1 to 7 describe the aerosol scenarios of gradually increase of scattering properties with a fixed $\sigma_{abs}$ of 0.050 km$^{-1}$, which cause the growths in both extinction and SSA. Case 8 to 11 present another process of the gradually increase of haze with more absorbing aerosols under the condition that $\sigma_{sca}$ are fixed on 0.250 km$^{-1}$, which consequently result in an increase extinction but decrease of SSA.

*Table 1. Simulation-based correlation information between $O_4$ DSCDs at 360.8 and 477.1 nm under conditions with different aerosol optical properties.*

| Aerosol information | | | | | Slope | $R^2$ | Intercept |
|---|---|---|---|---|---|---|---|
| No. | $\sigma_{abs}$ | $\sigma_{sca}$ | $\sigma_{ext}$ | SSA | | | |
| 1 | 0.050 | 0.075 | 0.125 | 0.6000 | 0.9560 | 0.9968 | 0.5516 |
| 2 | 0.050 | 0.125 | 0.175 | 0.7143 | 0.9117 | 0.9859 | 0.3178 |
| 3 | 0.050 | 0.250 | 0.300 | 0.8333 | 0.8089 | 0.9087 | 0.1438 |
| 4 | 0.050 | 0.350 | 0.400 | 0.8750 | 0.5672 | 0.8842 | 0.3861 |
| 5 | 0.050 | 0.500 | 0.550 | 0.9091 | 0.5649 | 0.9800 | 0.2305 |
| 6 | 0.050 | 0.700 | 0.750 | 0.9333 | 0.4519 | 0.9447 | 0.2603 |
| 7 | 0.050 | 1.000 | 1.050 | 0.9524 | 0.4875 | 0.9979 | 0.1654 |
| 8 | 0.010 | 0.250 | 0.260 | 0.9615 | 0.6754 | 0.7963 | 0.3948 |
| 9 | 0.025 | 0.250 | 0.275 | 0.9091 | 0.7682 | 0.9138 | 0.2353 |
| 10 | 0.075 | 0.250 | 0.325 | 0.7692 | 0.8051 | 0.9007 | 0.1407 |
| 11 | 0.100 | 0.250 | 0.350 | 0.7143 | 0.8446 | 0.9063 | 0.1011 |

Then, we did the linear-regression analysis for the simulated UV and Visible $O_4$ DSCDs under different aerosol conditions. As shown in Figure R1, the slope and $R^2$ between UV and Visible $O_4$ DSCDs illustrate that:

(1) Case 1-7 show an exponential trend in Figure R1. The fitting slope decrease accompanied with the increase of extinction coefficients and SSA if the condition of absorption coefficients are determined.

(2) Case 8-11 show a linear trend in Figure R1. The fitting slope will decrease

together with the decrease of extinction coefficients and the increase of SSA when the condition of absorption coefficients are determined.

(3) The correlation coefficients are high ($R^2$ are mainly greater than 0.90) for all the simulation results. As shown in case 8-11, $R^2$ decrease accompanied with the decrease of the correlation slopes. This conclusion need to be further supported by more detailed simulations.

[Figure]

*Figure R1. Correlation information (fitting slope and $R^2$) of the linear regression analysis between the simulated $O_4$ DCSDs at 360.8 and 477.1 nm under conditions of different aerosol optical properties in the simulation sensitivity studies.*

The forward RTM simulation results could demonstrate that the $O_4$ absorptions (the value of UV and Visible $O_4$ DSCDs, the corresponding linear-regression slope and $R^2$ between them) could greatly reflect the variation of aerosol optical properties, which present the theoretical evidences to some extent and enhance the principle basis of the proposed method. Moreover, the simulation results are consistent with the conclusions in the manuscript. The more detailed simulations in the future could provide the better quantitative relationship to the aerosol properties even more.

In addition, we have added this section to Discussion and the Supplement. Please refer to 333-339 in the manuscript.

Minor comments
(1) P4, Sect. 2.2 & P5, Fig. 1 => Please provide the basic information of the measured spectrum in the fitting example, which can be help to evaluate the performance of spectral analysis better.
R: The Fig. 1 in manuscript presents the typical spectral fitting of $O_4$ DSCDs in UV and Visible bands, and the corresponding measured spectrum were collected at 09:57:29 (SZA (Solar Zenith Angle) = 66.67º, SAA (Solar Azimuth Angle) = 48.40º, ELE (Elevation Angle) = 10º) and 09:42:29 (SZA = 68.27º, SAA = 49.29º and ELE = 5º) on 22 November 2016, respectively. We have supplemented the measured spectra information in the Fig. 1.
Besides, the completed spectral fitting were shown in Fig. R2 here.

[Figure]

*Figure R2. Example of DOAS spectral fitting of $O_4$ DSCDs in UV (left) and Vis (right) band. Black lines represent the absorption signal and red lines represent the sum of the absorption signal and the fit residual. The example spectrum used to retrieve UV and Visible $O_4$ DSCDs were obtained at 09:57:29 and 09:42:29 on 22 November 2016, respectively.*

(2) P8, L161-165 => Any special consideration for using different RH for clear days, non-haze days (RH < 80%) with haze days (RH ⩽ 80%)?

R: We apologized for this mistake of typing. In fact, we use RH < 80% to determine haze and heavy-haze days. In other words, RH all should be < 80% for clear, non-haze, haze and heavy-haze days. We have corrected this in the revised manuscript.

(3) P10, Fig.3 => Besides the discussion about the correlation coefficient, could the authors give some explanations of the changes in slopes among different weather types? Obviously, the slope in clear and non-haze days are much larger than those in haze and heavy-haze days. Why?

R: The oxygen collision complexes $O_4$ vertical profiles is well known and nearly constant in the atmosphere, the observed $O_4$ absorption can serve as an indicator for the atmospheric distribution of photon paths (Wagner et al., 2004; Frieß et al., 2006). The $O_4$ differential slant column densities (DSCDs) measured by MAX-DOAS are mainly attributed to the photon paths. Since the existence of aerosol can change the light path a lot, the variation of aerosol vertical profiles will be the main factor influencing the photon paths in a cloud-free sunny day, which will be further reflected in the observed $O_4$ DSCDs.

The path lengths from the effective scattering event to the telescope are dependent on

wavelengths. The path length in visible ranges is obviously longer than that in ultraviolet ranges. In clear and non-haze days, the path length is slightly affected by aerosols. However, the significant increasing of aerosol extinction coefficients in haze and heavy-haze days will have a large effect on the reduction in light path lengths. The reduced light path lengths are thought to result in small $O_4$ DSCDs. If these light path lengths are sufficiently shortened to penetrate hazy atmosphere, the measured $O_4$ DSCDs have large uncertainties and may lose sensitivity to vertical distributions of aerosol load (Lee et al., 2011). That could be the reason for the correlation slope in clear and non-haze days are much larger than those in haze and heavy-haze days.

Moreover, as shown in Figure R3, the simulation results could also greatly told us the correlation information (fitting slope and $R^2$) between $O_4$ DCSDs at 360.8 and 477.1 nm could changes under conditions with different aerosol optical properties. We also added the aerosol information on four different weather conditions in Figure 3 of the manuscript. The measured results and simulation results show good consistency on these four weather conditions of clear periods, light-haze periods, haze periods and heavy-haze periods, especially for heavy-haze periods.

[Figure]

*Figure R3. Simulation sensitivity studies of the correlation information (fitting slope and $R^2$) between $O_4$ DCSDs at 360.8 and 477.1 nm under conditions with different aerosol optical properties.*

(4) P12, Fig.4 => The scat. and abs. changed around 09:05 and 12:00, while the correlation relationship analysis use the break point of 10:00 and 12:00. Why they are different in time? Moreover, why the authors choose the index of variations of scat. instead of abs.?

R: The Figure 4 in the manuscript, we could find the scat. and abs. have lowest values at 09:05 during the time periods of 08:00 to 11:00. However, we have more focused on the change rate ($v_{sca}$, $a_{sca}$ and $a'_{sca}$ as defined in the manuscript) of scattering and absorption coefficients. The change rate of scattering coefficients ($v_{sca}$, $a_{sca}$ and $a'_{sca}$) could be better to help us to understand the relationship the $O_4$ DSCDs at different wavelength bands and the variations of $\sigma_{sca}$ and $\sigma_{abs}$. For example, the change rate of scattering coefficient at 10:00 is larger than that at 09:05.

We also try to choose the variation of absorption coefficients to identify the break point, but we found it could not identify all the break points as good as the variation

of scattering coefficients. Therefore, we choose the index of variation of scattering coefficients instead of absorption coefficients.

(5) P13-14, Fig.5&6 => The empirical relationships between measured O4 absorptions in different bands and characteristics of AOPs were mainly concluded from the statistic plot of Fig.5 and Fig.6. I have concerned that the some of the factors (e.g. correlation R2 and VIS/UV in haze days, as well as scat. and abs.) have wide value range even cover some cases of other weather conditions. How to obtain the precise and accurate the correspondence between O4 absorptions and AOPs under different weather conditions?

R: Thanks for your kindly suggestions. As shown in Figure R1 and R3, the fitting slope and $R^2$ have different values under conditions with different aerosol optical properties (scattering and absorption coefficient and the corresponding SSA information). Moreover, the corresponding values of $O_4$ DSCDs in UV and Visible ranges are also different under different conditions. Therefore, it will be a joint decision based on the values of $O_4$ DSCDs in UV and Visible ranges and the fitting slope and $R^2$ between them. This will help us to accurate the correspondence between $O_4$ absorptions and AOPs.

(6) P16-17, Sect.4 => For the validation, the authors classified the observational period segment into the different weather conditions, however, no further AOPs information, e.g., ADOs, scat. and abs., were inferred and achieved. Is the sentence in line 321("The $\sigma_{scat}$, $\sigma_{abs}$ and AOD are mainly located at 200-900 Mm-1, 20-60 Mm-1 and 0.9-2.5 under haze and heavy-haze conditions, respectively.") a conclusion of measurement results or inference from O4 absorptions?

R: We are very sorry that the description may cause some misunderstanding. The description has been updated as following:

Furthermore, the time series of in-situ $\sigma_{sca}$, $\sigma_{abs}$ and MAX-DOAS retrieved AOD are shown in Fig.7 (c) and (d). According to the empirical relationships summarized above, the $\sigma_{sca}$, $\sigma_{abs}$ and AOD should be mainly located at 200-900 Mm$^{-1}$, 20-60 Mm$^{-1}$ and 0.9-2.5 under the haze segment of 09:00-11:00 of 25 November. Simultaneously, the in-situ measured $\sigma_{sca}$, $\sigma_{abs}$ and MAX-DOAS retrieved AOD during the above same periods are ranged in 588.30-730.77 Mm$^{-1}$, 58.19-67.63 Mm$^{-1}$ and 1.39-2.22. The inferred results are in good agreement with the measured results. It indicates that the concluded empirical relationships can be used as the criterion to accurately determine the ranges of aerosol optical parameters of $\sigma_{sca}$, $\sigma_{abs}$ and AOD.

We have also supplemented this information in Line 323-329 in the manuscript.

Technical comments
(1) P2, L34: need to be developed
R: Please refer to Line 34.
(2) P2, L83: February of which year?
R: It should be February 2017. Please refer to line 82.

(3) P6, L143: growth => increase
R: Please refer to Line 14-141.

**Reference**

Frieß, U., Monks, P. S., Remedios, J. J., Rozanov, A., Sinreich, R., Wagner, T., and Platt, U.: MAX-DOAS O4 measurements: A new technique to derive information on atmospheric aerosols: 2. Modeling studies, J. Geophys. Res., 111, D14203, doi:10.1029/2005JD006618, 2006.

Lee, H., Irie, H., Gu, M., Kim, J., and Hwang, J.: Remote sensing of tropospheric aerosol using UV MAX-DOAS during hazy conditions in winter: Utilization of O4 Absorption bands at wavelength intervals of 338–368 and 367–393 nm, Atmospheric Environment, 45, 5760-5769, 10.1016/j.atmosenv.2011.07.019, 2011.

Wagner, T., Dix, B., von Friedeburg, C., Friess, U., Sanghavi, S., Sinreich, R., and Platt, U.: MAX-DOAS O4 measurements: A new technique to derive information on atmospheric aerosols–Principles and information content, J. Geophys. Res., 109, D22205, doi:10.1029/2004JD004904, 2004.

---

## Author Comment (AC2) · 28 Apr 2019

First of all, we would like to thank the reviewer for this positive assessment of our manuscript, the constructive and helpful suggestions. Point-to-point responses are given below. The original comments are black in color, while our responses are in blue. All the mentioned line number are referred to the revised manuscript.

Major criticism

(1)The approach to infer aerosol loading condition from the O4 absorptions is too far from quantitive, although authors mentioned to develop a lookup table to retrieve aerosol optical properties in the future. Aerosol loading information in the level of conditions defined in this study (clear, non-haze, haze, heavy-haze, ect) can be easily told from human eyes. Therefore, the aerosol information inferred from the method of this study has no scientific value.

R: Thanks for your comments. As shown in Figure R1, the retrieval of aerosol extinction coefficients profile from MAX-DOAS measurements is usually performed based on Optimal Estimation Method (OEM) or look-up table method in the previous research (the green box) (Frieβ et al., 2016; Beirel et al., 2019). Consequently, the yielded information is the vertical profile of aerosol extinction coefficient without any further specific scattering or absorbing coefficient. However, the research of this manuscript (the red box) is mainly concentrated on the identification of absorption and scattering coefficients directly from the $O_4$ absorptions with the empirical look-up table, rather than the aerosol loading information. In this way, the a priori information of aerosol for RTM simulation can be avoided, which contains considerable uncertainties in current OEM and look-up method.

[Figure]

***Figure R1. Aerosol profile inversion strategy.***

In order to enhance the principle basis of this proposed method, we have used radiative transfer model of SCIATRAN to simulate $O_4$ DSCDs in UV and Visible bands under conditions with different aerosol optical properties as a validation for the established empirical relationship. As listed in Table 1, 11 different aerosol scenarios were simulated in total, in which case 1 is the default case to represent haze condition. Case 1 to 7 describe the aerosol scenarios of gradually increase of scattering

properties with a fixed $\sigma_{abs}$ of 0.050 km$^{-1}$, which cause the growths in both extinction and SSA. Case 8 to 11 present another process of the gradually increase of haze with more absorbing aerosols under the condition that $\sigma_{sca}$ are fixed on 0.250 km$^{-1}$, which consequently result in an increase extinction but decrease of SSA.

*Table 1. Simulation-based correlation information between O$_4$ DSCDs at 360.8 and 477.1 nm under conditions with different aerosol optical properties.*

| | Aerosol information | | | | Slope | $R^2$ | Intercept |
|---|---|---|---|---|---|---|---|
| | $\sigma_{abs}$ | $\sigma_{sca}$ | $\sigma_{ext}$ | SSA | | | |
| 1 | 0.050 | 0.075 | 0.125 | 0.6000 | 0.9560 | 0.9968 | 0.5516 |
| 2 | 0.050 | 0.125 | 0.175 | 0.7143 | 0.9117 | 0.9859 | 0.3178 |
| 3 | 0.050 | 0.250 | 0.300 | 0.8333 | 0.8089 | 0.9087 | 0.1438 |
| 4 | 0.050 | 0.350 | 0.400 | 0.8750 | 0.5672 | 0.8842 | 0.3861 |
| 5 | 0.050 | 0.500 | 0.550 | 0.9091 | 0.5649 | 0.9800 | 0.2305 |
| 6 | 0.050 | 0.700 | 0.750 | 0.9333 | 0.4519 | 0.9447 | 0.2603 |
| 7 | 0.050 | 1.000 | 1.050 | 0.9524 | 0.4875 | 0.9979 | 0.1654 |
| 8 | 0.010 | 0.250 | 0.260 | 0.9615 | 0.6754 | 0.7963 | 0.3948 |
| 9 | 0.025 | 0.250 | 0.275 | 0.9091 | 0.7682 | 0.9138 | 0.2353 |
| 10 | 0.075 | 0.250 | 0.325 | 0.7692 | 0.8051 | 0.9007 | 0.1407 |
| 11 | 0.100 | 0.250 | 0.350 | 0.7143 | 0.8446 | 0.9063 | 0.1011 |

The linear-regression analysis for the simulated UV and Visible O$_4$ DSCDs under different aerosol conditions were also listed in Table 1. As shown in Figure R2, the slope and $R^2$ between UV and Visible O$_4$ DSCDs illustrate that:

(1) Case 1-7 show an exponential trend in Figure R2. The fitting slope decrease accompanied with the increase of extinction coefficients and SSA if the condition of absorption coefficients are determined.

(2) Case 8-11 show a linear trend in Figure R2. The fitting slope will decrease together with the decrease of extinction coefficients and the increase of SSA when the condition of absorption coefficients are determined.

(3) The correlation coefficients are high ($R^2$ are mainly greater than 0.90) for all the simulation results. As shown in case 8-11, $R^2$ decrease accompanied with the decrease of the correlation slopes. This conclusion need to be further supported by more detailed simulations.

We also added the aerosol information on four different weather conditions in Figure R2. The measured results and simulation results show good consistency on these four weather conditions of clear periods, light-haze periods, haze periods and heavy-haze periods, especially for heavy-haze periods.

Finally, it can be expected that the vertical spatial-resolved of aerosol scattering and absorption information can be retrieved by using O$_4$ DSCDs at different elevation angles, although we could only obtain these information at surface using the O$_4$ DSCDs at elevation angle 1° now. The accuracy of the determination for aerosol optical properties can be improved when the look-up table is replenished and refined in the future.

[Figure]

*Figure R2. Simulation sensitivity studies of the correlation information (fitting slope and $R^2$) between $O_4$ DCSDs at 360.8 and 477.1 nm under conditions with different aerosol optical properties.*

(2)Physically, the study methodology is not as much as sound. The O4 absorption derived from the MAX-DOAS observations are indeed contaminated by aerosols. So in their method, authors intend to use an aerosol-impacted O4 absorption data to infer the aerosol properties. I think a better way to get information from the MAX-DOAS observation is to retrieve aerosol information (if there is aerosol information) along with the O4 absorption. Or from another perspective, to study the impact of aerosols to the retrieval accuracy in the O4 absorption.

R: As shown in Figure R1, the current methodology to retrieve aerosol extinction vertical profile based on MAX-DOAS observed $O_4$ absorptions is mainly based on the OEM inversion strategy as indicated in the green box of Fig. R1. Considering the uncertainties on the a priori information in the stagey and the single yielded extinction coefficient, our research is aiming to obtain more detailed aerosol optical properties (with absorption and scattering vertical profiles) directly from the $O_4$ absorptions without the RTM simulation (inversion strategy is described in the red box).

Since the oxygen collision complexes $O_4$ vertical profile is well known and nearly constant, the observed $O_4$ absorption can serve as an indicator for the atmospheric distribution of photon paths (Wagner et al., 2004; Frieß et al., 2006). Therefore, the retrieved $O_4$ differential slant column densities (DSCDs) at different elevations can provide information about the impact of aerosol scattering on photon paths. By combining measurements of the $O_4$ absorption with radiative transfer model simulations, ground-based MAX-DOAS has been used in previous studies to determine atmospheric aerosol vertical extinction profiles and optical depths (e.g., Irie et al., 2008, 2009; Li et al., 2010; Clémer et al., 2010; Hartl and Wenig, 2013; Hendrick et al., 2014; Vlemmix et al., 2015; Frieß et al., 2016). Moreover, the UV and Visible $O_4$ DSCDs are used to retrieve aerosol extinction information independently.

In previous, our group have carried out several researches to retrieve aerosol extinction profiles using $O_4$ DSCDs, which also show the good agreements with external simultaneous measurements, such as Lidar, tethered-balloon observations

(e.g. Xing, et al., 2017; Tan et al., 2018). However, there are still some uncertainties on the retrieval based on Optical Estimation Method (OEM). So we try to establish the new method using $O_4$ absorptions in UV and Visible ranges to directly get aerosol scattering and absorption information in the study.

(3)I am not convinced with the change speed, acceleration, and the change rate of the acceleration of diurnal scattering coefficient (equations 2-4) for describing the aerosol property change. These variables only make the trivial diurnal analysis more complicated.

R: In order to illustrate the rationality of using the change speed ($v_{sca}$), acceleration ($a_{sca}$) and the change rate of acceleration ($a'_{sca}$) of diurnal scattering coefficients, we selected two examples (February 21 and 11, 2017) to show the evidences.

The example described in Figure R3 is also discussed in the manuscript. We could find the scattering ($\sigma_{sca}$) and absorption coefficients ($\sigma_{abs}$) have significant variations based on the calculated $v_{sca}$, $a_{sca}$ and $a'_{sca}$ at ~10:00 and ~12:00, respectively, while the time-indicated $O_4$ DSCDs seems to be three segments with higher correlation coefficients by the break point of 10:00 and 12:00 (in Figure R3 (a) and (c)).

[Figure]

*Figure R3. An example day on February 21, 2017: (a) the correlations between $O_4$ DSCDs*

*at 360.8 and 477.1 nm. The colorbar represents time sequence. (b) shows the time series of aerosol scattering and absorption coefficients. The correlations information between $O_4$ DSCDs at 360.8 and 477.1 nm during 08:00-10:00, 10:00-12:00 and 12:00-16:00 were shown in (c). (d) to (f) shows the time series of $v_{sca}$, $a_{sca}$ and $a'_{sca}$ of scattering coefficients, respectively.*

Another case in Figure R4, there are no dramatic changes on $\sigma_{sca}$ and $\sigma_{abs}$ during the day. The $\sigma_{sca}$ decreased slowly in the morning, kept in a stable level during 13:00-15:30, and then increased fast, which presented with three segmental periods (Fig. R4(b)). If we calculated $v_{sca}$, $a_{sca}$ and $a'_{sca}$ simultaneously, there is only one break point at 15:30 (Fig. R4(d)-(f)). Consequently, the time-indicated $O_4$ DSCDs of these two segmental periods have higher correlation coefficients. It also indicates aerosol scattering and absorption properties could be mainly manifested in the variations of $O_4$ absorptions at different wavelength bands.

[Figure]

***Figure R4.** An example day on February 11, 2017: (a) the time series of $O_4$ DSCDs at 360.8 and 477.1 nm. (b) the correlations between $O_4$ DSCDs at 360.8 and 477.1 nm. The colorbar represents time sequence. (c) shows the time series of aerosol scattering and absorption coefficients. (d) to (f) shows the time series of $v_{sca}$, $a_{sca}$ and $a'_{sca}$ of scattering coefficients, respectively.*

It can be concluded from Fig. R3 and R4 that the variations on the linear regression analysis between $O_4$ DSCDs in UV and Visible ranges could better reflect the variations of aerosol scattering and absorption properties. Moreover, the values of $O_4$ DSCDs are mainly dependent on the light path lengths. It will be difficult to find the break points of scattering coefficients, if we only rely on the values of $O_4$ DSCDs. Therefore, the above three parameters (change speed, acceleration and the change rate of the acceleration) can help us find the break points of scattering coefficients more accurately, for example in the Fig. R4, the proposed three index can better find the break point than the variation of $\sigma_{sca}$ itself.

In addition, the calculation of the change speed, acceleration and the change rate of the acceleration are carried out automatically by the coded program, which will not become complicated in practical.

(4)The presentation quality needs improvement. The paper has many grammar issues. I try to catch them in the technical comments below.

R: Thanks for your kindly suggestions. We have improved the presentation quality in the manuscript.

Specific comments

(1)It seems different datasets have different sampling times. It is not clear based on what time length the data are aggregated and compared. Please clarify this.

R: The scattering and absorption coefficient data, the main analytical data, measured at Peking University Urban Atmosphere Environment Monitoring Station (PKUERS, 39.9892°N, 116.3069°E) all have the sampling time of 5 minutes. However, the scattering and absorption data, the data used for validation, measured at Gucheng, Hebei province (39.1382°N, 115.7163°E) have different sampling temporal resolution. The sampling of scattering and absorption coefficient are 1 minute and 1 hour, respectively. We have also clarified in the manuscript. Please refer to line 128 and line 306-307.

(2)Line 160 – 165: the weather conditions are called clear days, … rainy days. Does it mean that all the data are analysis with on a daily based? If yes, it may be not appropriate, because different weather conditions can happen within a day. If no, these categories should not be called xx days.

R: All the data are analyzed with the hourly averages instead of a daily based. We will change "xx days" to "xx periods".

(3)I would suggested change "non-haze" into "light-haze", as "non-haze" means clear. And this condition has an average AOD of 0.35; calling light-haze is more proper.

R: We have corrected them in the whole manuscript according to your suggestion.

(4)Why the elevation angle = 1 degree is chosen for O4 DSCD used in this study? Please clarify it in the article.

R: Most of scattering and absorption observations were carried out by in-situ instruments at the ground surface. The ground surface aerosol also attracts more attentions due to its closer interactions with ecosystem and human health. Meanwhile, the $O_4$ absorptions at elevation angle $1^o$ could well reflect the near-surface aerosol information. We only have the in-situ measurements for scattering and absorption coefficients at the ground surface at this moment. Therefore, this study is mainly to focus on the near-surface aerosol optical properties.

Although it is difficult to measure scattering and absorption vertical profiles using in-situ instruments, we are attending to use air ship-based in-situ instruments to measure aerosol scattering and absorption vertical profiles in the near future, which can provide the completed profiles of aerosol scatting, absorption and extinction coefficients. Therefore, we would like further to use $O_4$ absorptions at multiple different elevation angles to study aerosol scattering and absorption coefficients at different heights. It will also could help us to improve the lookup table.

Technical comments

Line 24: I don't understand the meaning of "O4 Differential Slant Column Densities (DSCDs) at UV and visible bands varied in the order of …". Do you mean the magnitude of their correlation coefficients decrease I the order of …?

R: No, here we mean that the absolute value of O4 Differential Slant Column Densities (DSCDs) at UV and visible bands all varied "in the order of …". The DSCDs are usually to denote the absolute value of the results of spectral analysis.

40: aerosols and Aerosol Optical => aerosol loading and Aerosol Optical

40 – 41: Different aerosols behave => Different aerosol types behave

43: heat the air contributing to => heat the air, and contributes to

44: profile causing => profile, causing

R: Please refer to Line 40-44.

48: Please note the AE often refers to Angstrom Exponent.

R: We will use AEC and AE represent Aerosol Extinction Coefficient and Angstrom Exponent in our manuscript, respectively.

49: Angstrom => Angstrom Exponent

51: atmosphere in vertical. => atmosphere.

52: SSA could represent => SSA is defined as

R: Please refer to Line 49-52.

55: the four general aerosol types of biomass burning aerosol, urban-industrial aerosol, dust aerosol and aerosol of marine origin are exhibiting => different aerosol types (such as biomass burning, urban-industrial, dust and sea-salt aerosols) exhibit

R: Please refer to Line 55-56.

63: pronounced SSA => SSA

R: Please refer to Line 61.

71: with maxima at => around
R: Please refer to Line 71.

73: O4 absorptions can be yielded by the DOAS method and further the aerosol vertical profiles at four different wavelength bands (xx). => O4 absorptions in four bands (xx, xx, xx and xx nm) can be estimated, and aerosol vertical profiles can be further derived.
R: Please refer to Line 73-74.

90: Science (CAMS => Science building (CAMS
R: Please refer to Line 89.

95: were used to cover => are used to cover
R: Please refer to Line 94.

93: was equipped with => is equipped with
R: Please refer to Line 96.

106: in UV => in the UV; in visible spectral interval => in the visible
R: Please refer to Line 105.

113: filtered => filtered out; measurements remains for the further discussion. => measurements remained
R: Please refer to Line 111-112.

Table 1: The second line is confusing; No asterisk mark can be found for the table footnote.
R: The asterisk mark at bottom of Table 1 has been removed. We have added the statements about $I_0$ correction in the text body. Please refer to Line 109-110.

124: several times only during the daytime and only works on non-rainy days => about every 15 minutes during non-rainy daytime.
R: Please refer to 123.

133: Please indicate the distance between Beijing Airport and your site.
R: Please refer to 133.

141: always appeared when PM2.5 concentrations increased obviously, and the corresponding AOD also have a significant growth. => coincided with significantly high PM2.5 concentration and high AOD.
143: have a significant growth => increases dramatically
R: Please refer to Line 140-141.

143: gray area => gray areas
143: particles pollution => particulate pollution
R: Please refer to Line 142.

150: decreased faster and declined to => decrease sharply to
150: during a shorter while => within a shorter period
R: Please refer to Line 150.

153: are up to => are
154: greater than => over; all the wintertime => the entire wintertime
R: Please refer to Line 153-154.

176: decreased about => decreased by about
R: Please refer to Line 175.

189: at UV => in the UV
R: Please refer to Line 188.

Figure 3: please indicate the time is Beijing Time (UTC+8)
R: We have made the corrections. Please refer to Figure 3 in the revised manuscript.

248: weather types => weather conditions
R: Please refer to Line 250.

255: on the O4 => for the O4
255: at UV band => in the UV band; at visible => in the visible
R: Please refer to Line 257-258.

306: weather type => condition
R: Please refer to Line 310.

325: table. to retrieve => table to retrieve
R: Please refer to Line 332.

330: at UV and visible wavelength bands => in the UV and visible bands
R: Please refer to Line 341.

335: heavy-haze days to. => heavy-haze condition
R: Please refer to Line 346.

341: correlation slope => linear-regression slope
R: Please refer to Line 352.

**Reference**

Beirle, S., Dörner, S., Donner, S., Remmers, J., Wang, Y., and Wagner, T.: The Mainz profile algorithm (MAPA), Atmos. Meas. Tech., 12, 1785-1806, https://doi.org/10.5194/amt-12-1785-2019, 2019.

Clémer, K., Van Roozendael, M., Fayt, C., Hendrick, F., Hermans, C., Pinardi, G., Spurr, R., Wang, P., and De Mazière, M.: Multiple wavelength retrieval of tropospheric aerosol optical properties from MAXDOAS measurements in Beijing, Atmos. Meas. Tech., 3, 863–878, doi:10.5194/amt-3-863-2010, 2010.

Frieß, U., Monks, P. S., Remedios, J. J., Rozanov, A., Sinreich, R., Wagner, T., and Platt, U.: MAX-DOAS O4 measurements: A new technique to derive information on atmospheric aerosols: 2. Modeling studies, J. Geophys. Res., 111, D14203, doi:10.1029/2005JD006618, 2006.

Frieß, U., Klein Baltink, H., Beirle, S., Clémer, K., Hendrick, F., Henzing, B., Irie, H., de Leeuw, G., Li, A., Moerman, M. M., van Roozendael, M., Shaiganfar, R., Wagner, T., Wang, Y., Xie, P., Yilmaz, S., and Zieger, P.: Intercomparison of aerosol extinction profiles retrieved from MAX-DOAS measurements, Atmos. Meas. Tech., 9, 3205–3222, doi:10.5194/amt-9-3205-2016, 2016.

Hartl, A. and Wenig, M. O.: Regularisation model study for the least-squares retrieval of aerosol extinction time series from UV/VIS MAX-DOAS observations for a ground layer profile parameterisation, Atmos. Meas. Tech., 6, 1959–1980, doi:10.5194/amt-6-1959-2013, 2013.

Hendrick, F., Müller, J.-F., Clémer, K., Wang, P., De Mazière, M., Fayt, C., Gielen, C., Hermans, C., Ma, J. Z., Pinardi, G., Stavrakou, T., Vlemmix, T., and Van Roozendael, M.: Four years of ground-based MAX-DOAS observations of HONO and $NO_2$ in the Beijing area, Atmos. Chem. Phys., 14, 765–781, doi:10.5194/acp-14-765-2014, 2014.

Irie, H., Kanaya, Y., Akimoto, H., Iwabuchi, H., Shimizu, A., and Aoki, K.: First retrieval of tropospheric aerosol profiles using MAX-DOAS and comparison with lidar and sky radiometer measurements, Atmos. Chem. Phys., 8, 341–350, doi:10.5194/acp-8-341-2008, 2008.

Irie, H., Kanaya, Y., Akimoto, H., Iwabuchi, H., Shimizu, A., and Aoki, K.: Dual-wavelength aerosol vertical profile measurements by MAX-DOAS at Tsukuba, Japan, Atmos. Chem. Phys., 9, 2741–2749, doi:10.5194/acp-9-2741-2009, 2009.

Lee, H., Irie, H., Gu, M., Kim, J., and Hwang, J.: Remote sensing of tropospheric aerosol using UV MAX-DOAS during hazy conditions in winter: Utilization of O4 Absorption bands at wavelength intervals of 338–368 and 367–393 nm, Atmospheric Environment, 45, 5760-5769, 10.1016/j.atmosenv.2011.07.019, 2011.

Li, X., Brauers, T., Shao, M., Garland, R. M., Wagner, T., Deutschmann, T., and Wahner, A.: MAX-DOAS measurements in southern China: retrieval of aerosol extinctions and validation using ground-based in-situ data, Atmos. Chem. Phys., 10, 2079–2089, doi:10.5194/acp-10-2079-2010, 2010.

Tan, W., Liu, C., Wang, S., Xing, C., Su, W., Zhang, C., Xia, C., Liu, H., Cai, Z. and Liu, J.: Tropospheric NO2, SO2, and HCHO over the East China Sea, using ship-based MAX-DOAS

observations and comparison with OMI and OMPS satellite data, Atmos. Chem. Phys., 18, 15387–15402, https://doi.org/10.5194/acp-18-15387-2018, 2018.

Vlemmix, T., Hendrick, F., Pinardi, G., De Smedt, I., Fayt, C., Hermans, C., Piters, A., Wang, P., Levelt, P., and Van Roozendael, M.: MAX-DOAS observations of aerosols, formaldehyde and nitrogen dioxide in the Beijing area: comparison of two profile retrieval approaches, Atmos. Meas. Tech., 8, 941–963, doi:10.5194/amt-8-941-2015, 2015.

Wagner T., Dix, B., von Friedeburg, C., Friess, U., Sanghavi, S., Sinreich, R., and Platt, U.: MAX-DOAS O4 measurements: A new technique to derive information on atmospheric aerosols–Principles and information content, J. Geophys. Res., 109, D22205, doi:10.1029/2004JD004904, 2004.

Xing, C., Liu, C., Wang, S., Chan, K. L., Gao, Y., Huang, X., Su, W., Zhang, C., Dong, Y., Fan, G., Zhang, T., Chen, Z., Hu, Q., Su, H., Xie, Z., and Liu, J.: Observations of the vertical distributions of summertime atmospheric pollutants and the corresponding ozone production in Shanghai, China, Atmos. Chem. Phys., 17, 14275– 14289, https://doi.org/10.5194/acp-17-14275-2017, 2017.

---

## Author Response (AR2)

**Responses**

During the review on the revised manuscript, we got the comments from editor that:

The paper needs a technical revision to add the following concerns raised by the second reviewer before it can be accepted.

"Authors have properly addressed most of the reviewers' comments. I have only one question be clarified by the authors. In the review response (to reviewer #2), authors conducted simulation based correlation analysis for DSCDs between UV and visible bands under different aerosol conditions. It is not clear how the model simulations are sampled for the calculation of slope, $R^2$, and intercept in Table R1 and Figure R2. For each aerosol condition, Is the correlation analyzed for simulated DSCDs of different times of the day (i.e., different solar view angles) or else?"

R: We really appreciated the reviewer's helpful comments to make the manuscript better improvements. So we have made a technical revision to clarify the simulation study and the linear regression analysis more clearly in the manuscript, and in the supplement. The complementary sentences are marked in blue color in the manuscript.